# Exploring the Experience and Determinants of the Food Choices and Eating Practices of Elderly Thai People: A Qualitative Study

**DOI:** 10.3390/nu12113497

**Published:** 2020-11-13

**Authors:** Chalobol Chalermsri, Sibylle Herzig van Wees, Shirin Ziaei, Eva-Charlotte Ekström, Weerasak Muangpaisan, Syed Moshfiqur Rahman

**Affiliations:** 1Department of Women’s and Children’s Health, Uppsala University, SE-751 85 Uppsala, Sweden; sibylle.herzigvanwees@kbh.uu.se (S.H.v.W.); shirin.ziaei@kbh.uu.se (S.Z.); lotta.ekstrom@kbh.uu.se (E.-C.E.); syed.moshfiqur@kbh.uu.se (S.M.R.); 2Division of Geriatric Medicine, Department of Preventive and Social Medicine, Faculty of Medicine Siriraj Hospital, Mahidol University, Bangkok 10700, Thailand; drweerasak@gmail.com

**Keywords:** food choice, elderly people, healthy diet, caregiver, Thailand

## Abstract

Over the past decade, Thailand has experienced a rapid increase in its elderly population. Many unfavorable health outcomes among elderly people are associated with nutrition. Nutrition in elderly people is affected by physical, mental, and social factors. This study explored the food choices and dietary practices among community-dwelling elderly people in Thailand from the perspective of both caregivers and the elderly people themselves. Six focus group discussions and six semistructured interviews were conducted in the Samut Sakhon Province of Thailand. Deductive thematic analyses were conducted based on the “food choice process model framework.” The results show that physical and mental factors and societal factors are important determinants of food choices. Moreover, a changing food environment and economic factors were found to affect food choices. Issues of trust in food safety and food markets were highlighted as growing issues. Therefore, fostering healthy food choice interventions that consider both environmental and societal aspects is necessary.

## 1. Introduction

Globally, there is rapid growth in the elderly population. The number of elderly people aged 65 years and over has increased from 6% in 1990 to 9% in 2019 [1]. Elderly people are considered to be a group vulnerable to adverse health outcomes. Many elderly people experience a reduction in their physical and mental abilities [2] and require assistance with their daily activities, such as providing food for eating [3]. Therefore, the thoughts and opinions of caregivers should be considered as an important part of elderly people’s lives.

Elderly people with poor physical and mental health are likely to suffer various unfavorable outcomes [4]. Good nutritional status is one of the key components for sustaining a healthy life. However, nutritional disorders, in terms of undernutrition and overnutrition, are a common health threat to elderly people. Malnourished elderly people also have an increased risk of morbidities, mortality, and excessive healthcare expenditure compared to well-nourished elderly people [4,5].

Food choice pertains to the choices people make on the basis of their thoughts, feelings, and preferences associated with food and eating [6]. Food choice is an indicator of nutrient intake. Limited food choices are associated with poor dietary intake, worsened physical functions, and a decline in overall health status among elderly people [7]. Food choice consists of several components, such as biological processes [8,9,10], psychological conditions [11,12], and sociocultural [13,14,15,16,17,18], economic [19,20], and environmental factors [21,22,23,24]. Significant research on food choices has explored the perceptions of those consuming the food [19,25]. However, due to elderly people’s dependence on informal caregivers for food preparation in Thailand, there has been a growing call for more research on the perspective of caregivers [26,27]. Consequently, this study not only explored the perspective of the elderly on their food choices but also included investigated the perspectives of caregivers in elderly people’s food choices. This study focused on informal caregivers because this group represents the main caregiver group for elderly people in Asian countries instead of paid or formal caregivers [3,28]. This study fills a further gap in that it provides evidence on elderly people’s food choices from a middle-income country. Although there have been a number of studies about food choices among elderly people, the majority of these studies were conducted in high-income countries [29]. This study offers a thorough understanding of food choices among elderly people in such a setting, namely Thailand.

### 1.1. Nutrition and Aging Society in Thailand

Thailand is a middle-income country in Southeast Asia with a large elderly population. It is estimated that the elderly population in Thailand will increase from 8.6 million in 2019 to 13.8 million by 2030 [1]. Most elderly Thai people receive daily care from their families [30], and several health threats facing elderly Thai people are closely associated with nutrition. Although Thailand has launched many policies to improve eating behaviors, the double burden of malnutrition alongside the coexistence of undernutrition and overnutrition constitutes a major public health concern in Thailand, especially among elderly people [31]. A study by Churak et al. confirmed that around 10% of elderly Thai people are underweight, while more than 42% are overweight [31]. Hence, in-depth information on how elderly people select their food is a crucial component in understanding the behaviors surrounding nutritional problems among this population. Consequently, the objective of this study was to explore the experiences and determinants influencing the food choices and dietary practices among community-dwelling elderly people in Thailand from both their caregivers’ perspectives and their own perspectives.

### 1.2. Theoretical Framework

This research used the “food choice process model” as a theoretical framework to help guide the data collection tools and inform the data analysis to identify the determinants of individual food choices among the elderly and their informal caregivers [32,33]. This model suggests that food choices can be divided into three main parts: life course, influences, and personal food system (Figure 1). This model highlights the nexus between an individual’s food choices and eating behaviors and their own previous experiences during important events in their lives [34,35]. Moreover, this model highlights a series of influences, such as their individual health [9,10,11,35], interpersonal relationships in society [15,36,37], food environment [20,22,23,24,38], and the economic system they live under [20,39]. A combination of experiences and influences creates an individual’s food choice values. Finally, people use their own personal food systems, which are individual mental processes used to manage each food choice value. For example, people prioritize and balance food choice values in their minds to construct their own food choices and eating practices. Food choices have changed over time due to changes in ecology, culture, socioeconomic conditions, and global food systems [33]. Hence, food choices usually vary from one person to another and between people of different age groups. Globalization has transformed individuals’ shopping behaviors. Supermarkets and convenience stores are becoming increasingly popular and have replaced traditional wet markets [40,41]. The food choice process model has shown the complex interaction of factors related to food choice. This framework is more holistic and focused on changing over the lifetime. Therefore, this model is more suitable for study among the elderly whose life experiences act as an important component of food choice.

## 2. Materials and Methods

This is a qualitative study. This approach was selected to allow for the exploration of people’s subjective perceptions and practices. This work presents an exploratory qualitative study in which two data collection methods were used: focus group discussion (FGD) and semistructured individual interviews. Focus group discussions were completed as an initial step because this method can provide rich information on people’s perceptions and feelings [42] group dynamics during such discussions can reveal important key thoughts and opinions [43]. Thereafter, semistructured interviews were performed to explore the themes that emerged in the FGDs [44]. A combination of FGDs and individual interviews was performed to intensify our understanding of the studied phenomena and elaborate on the relevant information [45].

### 2.1. Research Setting

The study took place in Samut Sakhon Province, Thailand. Samut Sakhon Province is located approximately 60 km from Bangkok, the capital of Thailand. The number of individuals aged 60 years and older living in this province was approximately 14% in 2017, whereas the national estimate of people over 60 years is 16.1% [46]. This province was selected because it represents a suburban setting, which allows the study of characteristics related to both urban and rural contexts. A suburban case study was selected for a variety of reasons. First, the nutrients in foods between urban and rural foods differ [47]. Second, the prevalence of underweight and overweight in urban and rural settings also differs. People in rural areas have a higher prevalence of underweight but a lower prevalence of overweight compared to urban areas [48]. Moreover, Mahidol University, Thailand, plans to establish a geriatric training center in this province, so the collaboration between the present authors and healthcare organizations facilitated data collection.

### 2.2. Participant Sampling and Recruitment

Participants for FGDs were recruited from primary care units (PCU) in Mueang District, Samut Sakhon Province. Six PCUs were selected from among the 24 PCUs in this province. Recruitment of participants took place during elderly people’s club meetings in each PCU by community health workers. Purposive sampling was conducted, wherein the inclusion criteria for elderly people included being over 60 years of age, residing in Mueang district, Samut Sakhon Province, and having the ability to communicate in Thai. People with any medical or mental conditions that affected their communication abilities were excluded from the study. The inclusion criteria for the caregiver group included being an informal caregiver of an elderly person attending a PCU in Mueng District and the ability to communicate in Thai regardless of age or sex. Older participants and caregivers were not from the same family. Participants were recruited until saturation was reached. Repetition of data occurred after the completion of six FGDs [49].

Recruitment of participants for in-depth interviews was completed following the completion of FGDs because the research team felt that these people would be able to add additional information in a preferably private context. Three interviewees were recruited: one for male elderly, one for female elderly, and one informal caregiver. Additionally, we interviewed three further participants (Interviewee 1: man aged 77 years; Interviewee 2: woman aged 66 years; Interviewee 6: female caregiver aged 49 years) who did not wish to participate in FGDs but were happy to contribute to the study.

Overall, 36 participants were involved in the six FGDs (two groups of older men, two groups of older women, and two groups of caregivers) with six participants in each group. Following the FGDs, six participants—two male elderly people, two female elderly people, and two caregiver groups—were selected for semistructured interviews. The basic information of the participants is presented in Table 1.

### 2.3. Data Collection

The FGDs were conducted at the primary care units (PCUs), and the semistructured interviews were carried out at the participants’ houses or at coffee shops between July and October 2018. The FGDs and semistructured interviews were conducted in the Thai language. Both the FGDs and the interviews were audio-recorded and transcribed verbatim in Thai. Translation to English followed by a back-translation into Thai and accuracy rechecking were conducted by another expert prior to coding. The field notes were used to ensure the details of the transcripts, including nonverbal communication. Participants in the FGDs were grouped into female elderly people, male elderly people, and caregivers because participants usually feel more comfortable and prefer the company of others who share the same characteristics [50]. Experts in global nutrition and qualitative research reviewed all the guides prior to data collection. The FGD and interview guides were guided by our theoretical model, the “food choice process model,” described above [32,33]. Consequently, we explored perspectives on the four influencing factors: physical and mental health, food environment, society, and economic system. We also explored changes in food preferences over the life course.

Besides the theoretical framework, individual interviews were guided by the information that emerged from the FGDs, such as participants’ eating practices regarding the consumption of ready-to-eat food, the relationship between food and culture or society, chemical contamination in foods, and foods that were compatible with their daily activities. The duration of each discussion session ranged from 65 to 85 min, whereas each interview lasted approximately 40–62 min.

### 2.4. Data Analysis

The NVivo data analysis software program (NVivo 12 pro, QSR International Pty Ltd., Daresbury, Cheshire, UK) [51] was used to facilitate data coding, analysis, and organization.

The qualitative analysis consisted of a deductive approach by drawing on the “food choice process model” framework [33]. All transcripts were coded twice and culminated in the thematic map (Figure 2). Themes were then created by drawing on Braun and Clark’s thematic analysis [52]. Data analysis was conducted by C.C., S.M.R., and S.H.v.W. All the data, codes, and initial themes were reviewed by three researchers. The categories and themes were agreed upon in the research group.

### 2.5. Ethical Considerations

This study followed the principles of consolidated criteria for reporting qualitative research (COREQ) [53]. This study was approved by COA, Thailand Human Rights Committee for Research on Humans at Siriraj Hospital’s Mahidol University (No. Si 430/2018). According to the Declaration of Helsinki, all participants gave their written (or thumbprint for) informed consent prior to data collection. The informed consent consisted of an explanation of the study, the potential risks and benefits of the study, the participants’ voluntary decision competency, and the disclosure of information.

### 2.6. Strategies to Enhance the Study’s Quality

To improve the quality of this study, researchers applied the guideline for the rigor of qualitative research [54]. This study was conducted by a multidisciplinary team. The first author is a Thai geriatrician with experience in conducting qualitative and research on nutrition in the elderly. Two other research assistants were general practitioners in Thailand with experience in researching elderly people in their communities. These researchers were thus familiar with the context and language. Other team members (non-Thai) have expertise in nutritional, community-based, and qualitative research. The presence of a multidisciplinary and diverse team can help reduce bias. Additionally, three types of triangulation were applied to increase the study’s trustworthiness. Firstly, for methodological triangulation, we combined two data collection techniques: FGDs and individual interviews. Secondly, investigator triangulation used multiple independent coders. Furthermore, the thematic map was discussed in the research team. Lastly, for theoretical triangulation, deductive analysis was used following a theoretical model that was used to enhance the study’s credibility [55,56].

## 3. Results

Six FGDs and six individual interviews were conducted. The sociodemographic data of the participants are presented in Table 2. Four themes and nine subthemes related to elderly people’s food choices and dietary practices emerged from the data. Figure 2 illustrates the thematic map.

### 3.1. Physical and Mental Health

#### 3.1.1. Recognizing the Importance of Healthy Food

Both caregivers and elderly people appeared to have a good sense of what constitutes healthy food and what the benefits are for their health. One participant shared an example of adopting a healthier lifestyle by reducing sweet and salty food consumption and drinking more water:


*“In the past, I was not drinking enough. Right now [I] am drinking at least 1.5 L a day… I cut down the sweet foods and avoid all the salty ones.” (Interview 2, female elderly person)*


Most of the participants mentioned the influence of aging and chronic diseases on their food choices. Older participants classified fish and vegetables as soft food that they often ate due to their impaired dental health:


*“My teeth are not so good, so I have to eat fish and vegetables.” (FGD3, male elderly person)*


Older participants and caregivers trusted physicians and nurses and followed their advice. Older participants reported that they changed their food choices and eating habits because of a physician’s advice:


*“If the physician prohibits me from eating [specific foods] because of some diseases, then I have to stop eating it. There are so many foods that I could not eat such as fried-food, which is my favorite thing, but now, I have to eat less.” (FGD3, male elderly person)*


Caregivers aim to prepare nutritious and diverse foods for their care recipients, regardless of the elderly people’s opinions. However, some caregivers preferred to negotiate with the elderly people. One caregiver changed the protein source being served when her mother became bored and explained the benefits of healthy foods when she refused such foods:


*“My mother probably gets bored because of eating chicken and pork all the time. But I try to keep on changing [the food]. Sometimes she just wouldn’t eat at all. I try to convince her how good it is.” (Interview 3, caregiver)*


Despite the need to negotiate, the data suggest that there is a good understanding of the importance of healthy foods among both the elderly people and caregiver groups.

#### 3.1.2. Being Lonely

Loneliness was an important determinant of eating practices. Older participants complained that their loneliness and stress affect their eating practices. Almost all older participants have retired from their formal work, which leads to fewer opportunities to meet friends. A retired teacher said that food was not important when she lived alone.


*“If [I] am alone, [I] will only eat when [I] am hungry. If [I] don’t feel hungry, [I] will sleep. Sometimes [I] will sleep and skip a meal.” (Interview 2, female elderly person)*


Not feeling hungry and skipping meals are clearly described in the quote as a consequence of loneliness.

### 3.2. Society

#### 3.2.1. Eating with Family and Friends

Companionship during meals was an important determinant for the elderly participant’s positive eating habits. Elderly participants lived with their spouses during the weekdays, and their children visited them during the weekend. They often rated their family dinners as special events and noted that they were willing to cook for their family members.


*“Once the whole family gathers, we have a meal together. Like this weekend, my daughter called, “I’ll be here today. [I] will come to see mom and dad.” I said, “OK. What do you want to eat?” Then I also invited [my] grandchildren, the two [two grandchildren] in Bangkok. [They] told me what they wanted, and [I] will cook for them.” (Interview 1, male elderly person)*


Socializing with not only family but also friends has a positive impact on eating practices. Both older participants and caregivers indicated that they changed their food choices from buying ready-to-eat food to cooking for themselves when they had a meeting with their friends.


*“I usually buy ready-to-eat foods. But if I invite my friends, I will cook. I make red curry, bamboo curry with Yanang leaves [name of vegetables; Tiliacora Triandra], and papaya salad for the party… If I have [cooked] a lot of food, I will share it with the neighbors… I will cook and call the neighbors to eat together.” (FGD 5, caregivers)*


As the quote suggests, cooking for friends and neighbors also improved social relationships through the sharing of foods.

#### 3.2.2. Preferring Foods from One’s Own Ethnicity

Ethnicity appears to have an effect on an individual’s food preferences. This includes a preference for recipes from one’s own ethnic group and a preference for purchasing foods from people from their own ethnic group. Participants expressed concern over the cleanliness of migrant workers from other countries and preferred to purchase food from Thai sellers:


*“Migrant workers’ lifestyles are not hygienic, not very clean. Diseases that migrant workers have gets into the water, and it goes around. Some of them are food sellers. I have never bought food from migrant workers. I am afraid to buy food from migrant workers. I usually choose a stall that is owned by Thai people, not from neighboring countries’ people. I prefer the food to be made by Thai chefs and sellers.” (Interview 3, caregiver)*


The quotes suggest that ethnicity could affect food preferences. They strongly believe that people from their own ethnic group have a more hygienic lifestyle and provide safer, cleaner foods. Thus, people’s sociopolitical beliefs appear to be strongly connected to their individual food choices.

#### 3.2.3. Religion and Other Belief Systems

Beliefs and religion further influence food choices. All participants were Buddhists and avoided eating meat as much as possible because they followed the five precepts of Buddhism, which involve abstinence from killing humans or animals. Thus, they switched from animal products to plant-based products for their foods or decreased the amount of meat they consumed.


*“I don’t eat meat… we avoid [eating meat] because… Thai people are afraid of sins.” (FGD 2, male elderly person)*


Besides religion, local beliefs also influence food choices. For example, one elderly person believed that luffa in foods could increase milk production in lactating women.


*“In the past, there was a person who cooked spicy mixed vegetable soup for women after giving birth to increase milk. Luffa, fruits, and others were put in the soup.” (FGD 2, male elderly person)*


This participant was encouraged to eat traditional foods in which some herbs, such as luffa, were added because he believed in the benefits of adding herbs in such foods.

### 3.3. Food Environment

#### 3.3.1. Changing Availability of Food

Participants described changing food options as a result of changes in biodiversity. Both older participants and caregivers complained about the decline of the availability of natural species in their areas in terms of variety and quantity. For example, older participants discussed the changes in the characteristics of fish found at the local market.


*“The size of snappers used to be like this [gestured hand to estimate the fish size around 12 inches], but now we can’t even find baby snappers.” (FGD 2, male elderly person)*


Participants stressed the importance of seasonal variation. Many foods were available only during a specific time. One female elderly person mentioned a fruit that she could find only in the winter season.


*“Like if [I] want to eat a strawberry, [I] would choose to have it during the winter. Also, if [I] go to Chiang Mai [a city in the North of Thailand], [I] eat lots of them.” (Interview 2, female elderly person)*


Participant’s memories of food options from when they were younger were described to highlight significant differences in biodiversity. Older participants described the dwindling availability of natural food items compared to the past.


*“Marum [Moringa] sour soup is a food that we eat, the ingredients used to grow naturally. Now everything needs cultivation. Now there is no shrimp in the water.” (FGD 1, female elderly person)*


#### 3.3.2. Food Safety and Trust

Participants expressed major concerns about food safety. Both elderly people and caregivers were concerned about health-threatening conditions from contaminated foods:


*“It’s terrible… toxic substances. In the past, vegetables used to be watered naturally. Now farmers use insecticides and growth accelerator agents. For example, watermelon has a red-coloring accelerator agent.” (FGD 1, female elderly person)*


In addition to direct toxic substances, there were concerns about the effects of water pollution on food quality. Participants described problems with the wastewater from factories in the area. They believed that the decrease in the amount of fish was linked to this pollution:


*“Nowadays, some fishes are difficult to find because of the wastewater from the industry, which causes the fish to die. The majority of the small fishes die.” (FGD 3, male elderly person)*


Consequently, trust in food safety can affect food choices. Most of the participants and caregivers purchased foods only from trusted food providers, such as grocery stores or restaurants. They believed that foods from such sources were of better quality than foods from other sources.

### 3.4. Economic System

#### 3.4.1. Affording Food

Although individual economic circumstances affect older participants’ food choices to varying degrees, most older participants relied on a government pension for their living. Thus, the cost of food strongly influenced their food choices. Although most preferred high-quality food, they were limited by their financial status.


*“I want to eat good food but I can’t... The financial condition is the big issue.” (FGD 2, male elderly person)*


Financial difficulties are further highlighted by the fact that elderly people have to work, which, in turn, results in poorer quality food choices, such as convenience food.


*“I am a trader, so I don’t have time to cook foods or even boil eggs. If I cook curry and then go out to sell goods, by the time I return home, the curry is dry.” (FGD 2, male elderly person)*


#### 3.4.2. The Growing Popularity of Ready-to-Eat Foods

The pressure to work longer hours is perpetuating a ready-to-eat food culture. A participant shared her concerns over the changing and unhealthy items in food markets:


*“There are no nutritional [foods sold at] markets; all of them are (full) of chips and sugary drinks...” (Interview 2, female elderly person)*


Older participants rated ready-to-eat foods as their main food choices because cooking by themselves was a costly and time-consuming activity.


*“I would buy some ready-to-eat food from the market, heat it up, and eat with the family. We rarely cook because buying a cooked meal is very convenient. To cook a meal costs quite a lot of money. I also don’t want to spend too much of my time on cooking.” (Interview 5, male elderly person)*


Caregivers also had the same experiences. One caregiver said that she takes care of her mother, runs her own small business, and acts as a health volunteer at the same time. She described cooking as a time-consuming activity. Thus, she bought ready-to-eat foods from local markets for her mother.


*“Right now, [I] don’t have any time to cook. [I] need to keep up with the schedule, so [I] have to buy ready-to-eat foods.” (Interview 3, caregiver)*


## 4. Discussion

The aim of this study was to understand elderly people and their informal caregivers’ views on food choices in the context of their communities in Thailand. This study was framed by the theoretical “food choice process model” and explored four types of factors that are known to affect food choices: individual factors (physical and mental health), social factors, environmental factors, and economic factors.

In particular, recognizing the importance of foods for their personal health had an effect on elderly people’s decisions to eat healthy foods. Loneliness affected the amount and types of foods consumed. Interpersonal relations within a participant’s family and society also modified food choice. Environmental and economic systems also appeared to influence an individual’s food choices. Both elderly people and caregivers recognized the importance of healthy foods and adapted their dietary practices to comply with elderly people’s health conditions. These findings are consistent with those of previous studies showing that elderly people select foods that help them achieve a healthy life [9]. For example, elderly Dutch people consume healthy food to maintain their physical fitness, stabilize their health status, and prevent chronic diseases, especially those of high educational and financial status [10]. Other studies compared the dietary practices between the older and younger population, showing that elderly people consume more healthy foods [57,58]. Besides concerns over their health, these individuals also have nutritional knowledge. For example, some limit their egg consumption due to the cholesterol in yolk [59]. Conversely, many studies have shown that elderly people do not pay attention to the health aspects of foods but that such individuals, instead, tend to rely on their previous life experiences. Irish elderly people prefer to eat “simple” foods such as bread with butter or jam, eggs, and milk [60]. Another study demonstrated that elderly people often eat animal’s internal organs and yolks because they were habituated to eating such foods since childhood, regardless of their nutritional benefits [59].

Besides physical health, mental health also affects elderly people’s food choices. Elderly people’s loneliness resulted in limited food intake. Previous studies have shown that living and eating alone is strongly associated with loneliness [35]. A study on elderly people from the United Kingdom observed that loneliness and eating alone produces a lack of cooking and motivation to eat [11,35]. Elderly people also mentioned that they shop and cook for themselves and, thus, usually eat easy-to-make foods [11]. Besides feeling alone, older bereaved women reported a decrease in their appetites and motivation to cook [35]. Although loneliness and living and eating alone have been commonly mentioned as determinants of food choices among elderly people, the impacts of these determinants on an individual’s health remain controversial. Previous studies have shown that elderly people who eat alone have a lower quality of life, less food variety, a lower BMI, and a higher risk of malnutrition than people who eat with others [12,61,62]. However, a study from Finland showed that only the feeling of loneliness is related to elderly people’s nutritional status, while living alone is not itself a determinant [63]. Regarding interpersonal relationships, social eating with family and friends was commonly mentioned in this study. Elderly people seem to pay more attention to good cooking when they eat with their family members or friends. This result is similar to the findings from previous studies conducted among low-income independent elderly people in the United States, for whom social interaction was highlighted as an important determinant for cooking foods. These individuals preferred to eat meals at the center for elderly people rather than eat alone [34]. Another study among frail elderly people found that people who have close relationships with their families or friends usually depend on others’ advice for food choice selection [15]. Thus, interpersonal relationships also have effects on the participants’ thoughts and behaviors.

In this study, participants preferred to eat foods from their own ethnic local food providers. This finding is consistent with previous studies, as people from different ethnicities and cultures have their own food patterns [16]. A previous cross-cultural study exploring the behaviors between Thai and US customers showed that American people visit both Eastern and Western-style restaurants while Thai people prefer to eat only at Asian restaurants, such as Thai, Japanese, or Chinese restaurants. Besides eating out, Thai people prefer to shop at ethnic grocery shops more often than American people (98% vs. 49%) [17]. Such behaviors have been linked to the concept of food neophobia. Food neophobia is explained as the reluctance to eat and/or the avoidance of novel foods [64]. Elderly people present food neophobic behavior more often than younger groups, especially those with low education, low income, or those living alone [65,66]. Although food neophobia has seemingly increased among elderly people, the consequences of such behaviors have not been thoroughly explained.

In this study, both the elderly participants and their caregivers talked about/discussed/communicated their religious and nonreligious beliefs. This finding supports previous studies in this field. A study regarding women’s perspectives on foods in Iran highlighted the importance of religious values as influencers of food choices [18]. Another study in India found that Hindu families who follow their religion strongly are vegetarians or consume meat less frequently than other groups [14].

Changes in the providing environment was another key factor that affected the food choices of the participants, who complained that their food intake was affected by diminished biodiversity. This finding is compatible with previous studies. A previous qualitative study revealed that reduced biodiversity and decreases in the variety of available agricultural products are barriers to the consumption of traditional foods [22]. Furthermore, another study among low- and middle-income countries showed a positive association between both biodiversity and food diversity and nutrient adequacy [23]. The continuous changes in the climate, moreover, affect all food systems from production to consumption and are related to overall food insecurity and malnutrition [67].

In this study, ready-to-eat foods were the most popular among both elderly people and caregivers for economic reasons. Globally, the consumption of ready-to-eat food or convenience food has increased rapidly [68]. In a past study, elderly people in Finland felt that the price of ready-to-eat food was greater than that of self-made food because self-made foods can be eaten over several days [20]. However, it is difficult to compare the ready-to-eat food in this study with those of Western countries due to many different factors. Thai ready-to-eat foods are made by small, local food sellers, whereas in Western countries, ready-made foods are found in supermarkets, grocery shops, and restaurants [69]. Apart from direct economic reasons, ready-to-eat foods are preferred because they are time saving and easily accessible [69]. People can buy these foods from anywhere they want, such as markets, Sunday markets, restaurants, or on the street. This also accords with earlier observations. People aged 60–88 years in New York in the United States reported that they often bought foods from nearby grocery stores [21].

In the present study, food safety was a concern in all processes, from food production to retail. Participants worried about food contamination from insecticides or preservatives. This result matches the observed findings in earlier studies. More than half of Chinese customers are concerned about the chemical ingredients, pesticides, hormones, or antibiotics that might be present in the foods they eat [24]. Besides worrying about the chemical contamination in foods, hygiene practices and food cleanliness were also of great concern. A study exploring Chinese customer’s perceptions and awareness towards food showed that food safety is the highest of all concerns and includes food hygiene, food poisoning, and diseases that are transmitted through foods or result from chemical ingredients added into the foods [24]. In this study, participants believed that products from supermarkets were cleaner and safer than products from general food markets. This is in accordance with an earlier study showing that food safety is the primary issue that concerns customers visiting supermarkets and convenience stores [41].

### Strengths and Limitations of the Study

The strengths of this research are its inclusion of both elderly people and caregivers in a study setting that has received little attention in the literature. Because of their physical and cognitive limitations, many elderly people rely on their caregivers for food preparation and other general care. Hence, information from both groups can provide a more holistic perspective on elderly people’s nutrition and food preferences. The information from elderly people and caregivers in middle-income countries has received little attention in previous studies. Therefore, the present study helps fill this knowledge gap. Researchers adopted strategies to ensure the credibility of the research, conducted by the multidisciplinary research team. Different perspectives from the multidisciplinary team improved not only the comprehensiveness of the interview’s guide but also analysis and interpretation processes. Interviews guides were derived from a theoretical framework and reviewed by experts in the relevant fields. Three types of triangulation were applied in this study, methodological, investigator, and theoretical triangulation.

On the other hand, participants were recruited by community health workers at each PCU, which might have caused selection bias. People in contact with healthcare personnel and facilities had a higher probability of joining this study. Furthermore, the interviewer–interviewee relationship might have disturbed the participants’ responses. Participants knew that the researchers were medical personnel. Consequently, many participants attempted to demonstrate their knowledge and discuss the concepts in a socially desirable fashion. For example, some participants described food for diabetic patients rather than precisely describing the foods they themselves ate. Nevertheless, we stated that we wanted to know the participants’ real eating habits and emphasized that there were no right or wrong answers as all answers would be accepted. The study aimed to explore the experiences and determinants influencing the food choices and dietary practices from a specific setting rather than a generalization to the wider population. However, the results of this study are transferable to other populations with similar characteristics [70].

## 5. Conclusions and Implications for Policy and Practice

The “food choice process model” proved a suitable framework to explore the determinants and food choices of elderly people and their caregivers in Thailand. This study confirmed that the determinants of food choices and dietary practices, as described in the “food choice process model,” are multifactorial and include not only an individual’s physical and mental health but also society, the food environment, and economic systems. Elderly people pay attention to a healthy diet and food safety to maintain their health status. Social factors in terms of family, religion, and ethnicity are further influencing factors. To promote healthy eating habits among elderly people, their needs and requirements have to be identified. While some elderly people have positive eating habits, this is unlikely to be the case for all elderly people. Elderly people in this study knew of the importance of food for their health. However, their specific food preferences inhibited good food habits. For example, ethnic food preferences were a major concern for the elderly people interviewed, which arguably lowered their food availability. Understanding the implications of this dynamic in more detail would help improve our current understanding of food availability. Furthermore, this study found that ready-to-eat foods were popular among elderly people. Monitoring this change and learning more about it through research could help improve our current understanding of this pattern and the implications thereof. A further point raised in this study was people’s concern over food safety. The implications of decreasing trust in food markets and concerns over food safety may be a growing influencing factor of food choices, especially in current times since infectious diseases such as COVID-19 are likely to have originated in food markets. More research on the perceptions of these markets and the implications thereof could be helpful in understanding how these factors affect people’s dietary practices. A better understanding of food choices can contribute to devising evidence-based policies and health promotion interventions that address the poor nutritional status of elderly people in Thailand and possibly beyond.

## Figures and Tables

**Figure 1 nutrients-12-03497-f001:**
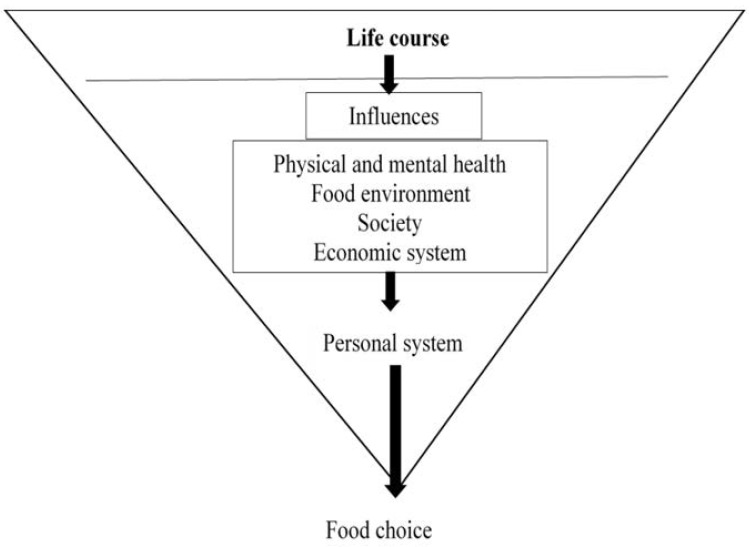
A food choice process model [32,33].

**Figure 2 nutrients-12-03497-f002:**
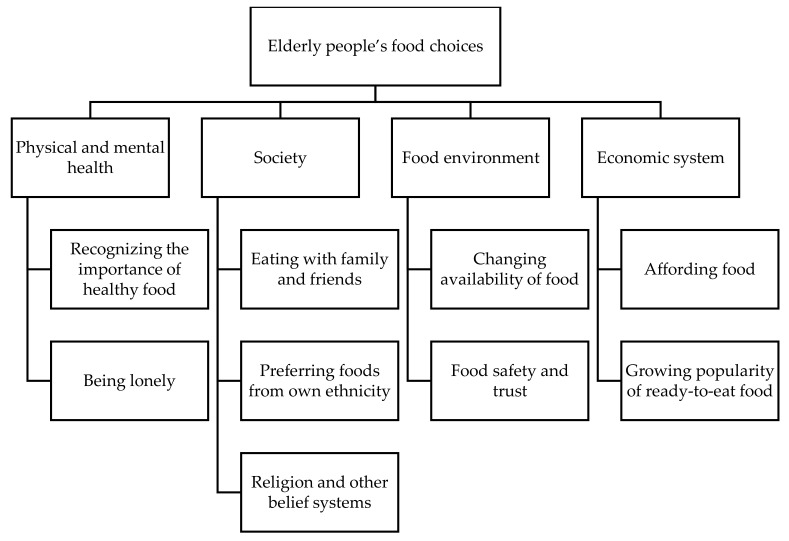
Elderly people’s food choices and dietary practices (adapted from the food choice process model).

**Table 1 nutrients-12-03497-t001:** List of participants.

FGD	Interviews
1: women aged 61–69 years	1: man aged 77 years
2: men aged 62–76 years	2: woman aged 66 years
3: men aged 69–81 years	3: caregiver 64 years
4: caregivers aged 56–72 years	4: woman aged 63 years
5: caregivers aged 51–71 years	5: man aged 65 years
6: women aged 64–76 years	6: caregiver aged 49 years

**Table 2 nutrients-12-03497-t002:** Sociodemographic data of the study participants.

Characteristics	Elderly People (*n* = 26)	Caregivers (*n* = 13)
Age (years); median (IOR)	71.5 (64.8–76.0)	64.0 (58.0–70.0)
Gender, male; *n* (%)	13(50.0)	2 (15.4)
Have formal education; *n* (%)	23 (88.5)	13 (100.0)
Marital status; *n* (%)		
Single	1 (3.8)	2 (15.4)
Married	18 (69.2)	8 (61.5)
Separated or widowed	7 (26.9)	3 (23.1)
Number of people in household including the individual; median (IOR)	3 (2.0–6.0)	4 (2.5–6.0)

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
