# Peer review of "Exploring the Experience and Determinants of the Food Choices and Eating Practices of Elderly Thai People: A Qualitative Study"

_nutrients, 2020, doi:10.3390/nu12113497_

Round 1

Reviewer 1 Report

The research topic was interesting, and I have some questions.

..

  1. As you mentioned about the coexistence of undernutrition and overnutrition constitutes a  major concern in Thailand, did you find any interesting issue correlated to individual’s food choice.

(For example; overweight elderly tended to select more healthier food or totally ignored about their food.)

  1. I wonder how did you selected the six participants for semi-structured interviews and how to match each of them to each FGD group.

  1. I found it in Table 1. that you clarified the education of participants into three categories, but you didn’t mention the influence of education to the food choice in either results and discussion.

Reviewer 2 Report

Overall, the paper is interersting an well-written and provides a rare insight into the food choices of Thai elderly population. However, the manuscript requires significant improvements before it can be published.

General remarks:

  1. In the Introduction, the authors do not provide enough background to justify their choice of the sample (e.g. why caregivers are included) and their theoretical framework. Moreover, it is not clear which research gap exists and what is the research question of the study. Some of the relevant literature is provided in the Discussion section of the paper. I would suggest to use some sources from the Discussion section for the literature background in the Introduction.
  2. It would be useful to discuss more the "food proces model", why it was chosen and if it was used for the qualitative research before.
  3. More information is required regarding the implementation of focus groups and individual interviews: which questions were asked, were there any differences between the focus groups and the interviews, how the questions correspond to the theoretical model?
  4. It is not clear how the information in the Results section answers the research question(s). Was the point of the study to confirm that that the food choice process model can be applied for a qualitative analysis? Or was the model adapted after the research was completed to structure the results? What was the point of the study then and what is the role of the model?
  5. Obvious but nontheless important limitations, like e.g. sample size and sample structure are not discussed. 

Minor comments: 

I suggest excluding section 2.6. It raises more questions than it justifies the methodology. 

Check the translation of participants' quotes - a lot of mistakes there. 

There is no number and title for the table in 2.2. Participant Sampling and Recruitment.

Shouldn't sections 3.1.2. Being Lonely and 3.2.1. Eating with Family and Friends refer to the same topic?

Round 2

Reviewer 2 Report

I agree with the publication of the manuscript in its present form.

Author Response

Thank you again for your comments.